# Activation of Interferon-Stimulated Genes following Varicella-Zoster Virus Infection in a Human iPSC-Derived Neuronal In Vitro Model Depends on Exogenous Interferon-α

**DOI:** 10.3390/v14112517

**Published:** 2022-11-14

**Authors:** Marlies Boeren, Elise Van Breedam, Tamariche Buyle-Huybrecht, Marielle Lebrun, Pieter Meysman, Catherine Sadzot-Delvaux, Viggo F. Van Tendeloo, Geert Mortier, Kris Laukens, Benson Ogunjimi, Peter Ponsaerts, Peter Delputte

**Affiliations:** 1Laboratory of Microbiology, Parasitology and Hygiene (LMPH), University of Antwerp, 2610 Antwerp, Belgium; 2Laboratory of Experimental Hematology (LEH), Vaccine and Infectious Disease Institute (VAXINFECTIO), University of Antwerp, 2610 Antwerp, Belgium; 3Antwerp Center for Translational Immunology and Virology (ACTIV), Vaccine and Infectious Disease Institute, University of Antwerp, 2610 Antwerp, Belgium; 4Laboratory of Virology and Immunology, GIGA-Infection, Inflammation and Immunity, University of Liège, 4000 Liège, Belgium; 5Antwerp Unit for Data Analysis and Computation in Immunology and Sequencing (AUDACIS), 2610 Antwerp, Belgium; 6Adrem Data Lab, Department of Computer Science, University of Antwerp, 2610 Antwerp, Belgium; 7Biomedical Informatics Research Network Antwerp (Biomina), University of Antwerp, 2610 Antwerp, Belgium; 8Department of Medical Genetics, Antwerp University Hospital, University of Antwerp, 2610 Antwerp, Belgium; 9Centre for Health Economics Research & Modelling Infectious Diseases (CHERMID), Vaccine & Infectious Disease Institute (VAXINFECTIO), University of Antwerp, 2610 Antwerp, Belgium; 10Department of Paediatrics, Antwerp University Hospital, 2610 Antwerp, Belgium; 11Infla-Med, University of Antwerp, 2610 Antwerp, Belgium

**Keywords:** varicella-zoster virus, neurons, interferon-α, interferon-stimulated genes, neuronal models, iPSC, innate immune response, antiviral response, axonal infection

## Abstract

Varicella-zoster virus (VZV) infection of neuronal cells and the activation of cell-intrinsic antiviral responses upon infection are still poorly understood mainly due to the scarcity of suitable human in vitro models that are available to study VZV. We developed a compartmentalized human-induced pluripotent stem cell (hiPSC)-derived neuronal culture model that allows axonal VZV infection of the neurons, thereby mimicking the natural route of infection. Using this model, we showed that hiPSC-neurons do not mount an effective interferon-mediated antiviral response following VZV infection. Indeed, in contrast to infection with Sendai virus, VZV infection of the hiPSC-neurons does not result in the upregulation of interferon-stimulated genes (ISGs) that have direct antiviral functions. Furthermore, the hiPSC-neurons do not produce interferon-α (IFNα), a major cytokine that is involved in the innate antiviral response, even upon its stimulation with strong synthetic inducers. In contrast, we showed that exogenous IFNα effectively limits VZV spread in the neuronal cell body compartment and demonstrated that ISGs are efficiently upregulated in these VZV-infected neuronal cultures that are treated with IFNα. Thus, whereas the cultured hiPSC neurons seem to be poor IFNα producers, they are good IFNα responders. This could suggest an important role for other cells such as satellite glial cells or macrophages to produce IFNα for VZV infection control.

## 1. Introduction

Varicella-zoster virus (VZV) or human herpesvirus 3 (HHV-3) is a human neurotropic alpha-herpesvirus belonging to the *Herpesviridae*. VZV naturally infects more than 95% of the population [1,2]. Upon the entry of VZV in the body, the infected T cells can transport VZV to the skin and cause varicella (chickenpox) which is usually self-limiting, although serious complications can occur [2,3]. The VZV particles may access sensory nerve endings in the skin and can be transported to the neural ganglia via retrograde axonal transport where VZV latency is established [2]. Alternatively, VZV can be readily transported to the neuronal cell bodies by the infected T cells, thereby establishing latency [4,5]. Experiments with simian varicella virus (SVV), the nonhuman counterpart of VZV, have already shown that SVV DNA can be detected in the ganglia before the appearance of skin lesions, thus favoring the lymphocyte vector for the infection of the neurons [6]. VZV can reactivate from this latent state, traveling back to the skin via anterograde transport, thus resulting in herpes zoster (HZ, shingles) [2]. A substantial part of HZ patients suffers from long-lasting pain after the rash has disappeared, i.e., postherpetic neuralgia (PHN) [7,8]. PHN is an important complication of HZ as it results in a decreased quality of life and a considerable economic cost from drug prescription and work abstinence [7,9]. VZV can also reactivate from the cranial nerve ganglia, causing HZ ophthalmicus, and from the ganglia in the cervical, thoracic, or lumbosacral region, which can cause weakness (zoster paresis) [10]. Current therapeutic treatment options for HZ include valaciclovir, famciclovir, and acyclovir. However, these antiviral treatments are only proven to be effective in reducing the severity of acute pain and the duration of the skin lesions when they are administered within 72 h after onset of HZ, and they do not prevent PHN [11,12]. Although effective vaccines against HZ are now available, the vaccination coverage worldwide is relatively low [13,14,15,16]. Hence, VZV and HZ will remain a major burden for the foreseeable future.

The VZV genome is composed of a large linear dsDNA strand that is approximately 125 kb [17]. The genomes are packed into a nucleocapsid which is formed by multiple proteins including the minor capsid protein which is encoded by ORF23 [18,19,20]. The nucleocapsid is surrounded by a tegument layer that is made up of proteins with regulatory functions including the VZV transactivating factor proteins ORF62 and ORF63 [20,21,22]. The nucleocapsid and tegument layers are enveloped by structures that are derived from cellular membranes with incorporated viral glycoproteins [2,20]. In latently VZV-infected human trigeminal ganglia (TG), the transcription is restricted to the VZV latency-associated transcript (VLT) and RNA 63, encoding ORF63 [23]. VLT maps antisense to ORF61, and shares properties with the classical LAT of HSV and other neurotropic alpha-herpesviruses [24]. VLT is, however, not only present during latency, but it encodes a protein with late expression kinetics in productively infected cells in vitro and in HZ skin lesions [24]. Furthermore, VLT-ORF63 fusion transcripts were identified in human TG, and they were found to be induced by reactivation stimuli, whereas VLT is transcribed during in vitro latency. One isoform of VLT-ORF63 induces broad viral gene transcription and is potentially involved in the transition from VZV latency to a lytic infection [23]. The experiments that are presented in this manuscript were carried out on in vitro cultured stem cell-derived neurons.

The innate immune response is indispensable for early VZV control. Type I interferons (IFNs), such as IFNα and -β, are induced at high levels upon the sensing of VZV via the pattern recognition receptors (PRRs) on innate immune cells and non-immune cells that recognize the pathogen-associated molecular patterns (PAMPs) [25,26]. Cell surface Toll-like receptor 2 (TLR2) recognizes the VZV particles, presumably via the viral glycoproteins [27]. Endosomal TLR9 recognizes the viral genomic dsDNA that is rich in unmethylated CpG DNA motifs [27], and endosomal TLR3 senses the VZV dsRNA [28]. Furthermore, herpesvirus’ viral dsRNA intermediates, and ssRNA with a 5′-triphosphate group are detected by the cytosolic RNA receptors melanoma differentiation-associated protein 5 (MDA5) and retinoic acid-inducible gene-I (RIG-I), whereas viral dsDNA is recognized by the DNA receptors such as RNA polymerase III (POL III) and cyclic GMP-AMP synthase (cGAS) [27,29]. cGAS detects the DNA in a sequence-independent manner and its enzymatic activity leads to production of 2′3′ cGAMP which activates the stimulator of the interferon (IFN) genes (STING) [26]. POL III can transcribe AT-rich DNA, such as the VZV genomic dsDNA, and generate 5′ triphosphorylated RNA species which are agonistic compounds for RIG-I. RIG-I activation in turn leads to signaling to the mitochondrial antiviral signaling proteins (MAVS). Both pathways (cGAS and POLIII) act via the downstream activation of interferon regulatory factory 3 (IRF3) and ultimately lead to the production of type I IFNs, including IFNα and IFNβ [26]. Type I IFNs can bind to IFNα receptor 1 (IFNAR1) and 2 (IFNAR2) and finally exert their effect through the induction of a large range of IFN-stimulated genes (ISGs) which have direct antiviral effector functions [30,31]. The four main IFN-mediated effector pathways which have been identified are: the protein kinase R (PKR) pathway, the 2′, 5′ oligoadenylate-synthetase-directed ribonuclease L (OAS-RNaseL) pathway, the interferon stimulated gene 15 (ISG15) ubiquitin-like pathway, and the Mx GTPase pathway [32].

The human-tropic nature of VZV significantly hampers our understanding of VZV infection and pathogenesis [33,34]. In fact, no small animal model so far has succeeded to evoke a response that is fully equivalent to that in humans [33,34]. Human cadaver ganglia have been studied to circumvent the problem of VZV’s human-tropic nature, but these also have considerable limitations due to their inherent variability, natural decomposition and, importantly, their limited availability [34]. During the past decade, neuronal in vitro models using human embryonic stem cells (hESC) or human induced pluripotent stem cells (hiPSC) have been used to study VZV infection [35,36,37,38]. Neurons that are derived from hESC and hiPSC can also be cultured in compartmentalized chambers that separate the axons from the neuronal cell bodies, thus allowing for axonal VZV infection which mimics the natural infection route [39,40]. Pioneering work from Markus et al. showed that hESC-derived neurons in compartmentalized chambers can support a productive VZV infection, as well as a quiescent infection [38]. In addition, hiPSC have the potential to create co-cultures of neurons with autologous (patient-derived) immune cells, and thereby, they will have the advantage to study more complex virus-host interactions.

Here, we described the development of a human iPSC-derived neural compartmentalized model of VZV infection that mimics the in vivo infection route via axon termini. We showed that while hiPSC neurons adequately respond to the INFα treatment by the production of ISGs, they do not produce detectable levels of IFNα mRNA upon their inoculation with VZV or Sendai virus (SeV), nor upon their stimulation with synthetic IFN inducers. In contrast, the hiPSC neurons produce ISGs upon their stimulation with the synthetic inducers poly(dA:dT) and poly(I:C), and upon the SeV infection, but not upon the VZV infection.

## 2. Materials and Methods

### 2.1. Neuronal Differentiation Protocol

A previously established iPSC line originating from human foreskin fibroblasts (ATCC, CRL-2522) and generated using CytoTune™-iPS 2.0 Sendai Reprogramming Kit (Thermo Fisher Scientific, Waltham, MA, USA) was differentiated into a self-renewable neural stem cell line (hiPSC-NSC) using PSC neural induction medium (Gibco^TM^, Evansville, IN, USA) [41]. Characterization of this hiPSC-NSC line included CNV sequencing, demonstrating the absence of major genetic abnormalities and the bi-lineage differentiation potential [41]. This hiPSC-NSC line was cultured on 6-well plates that were coated with Geltrex LDEV-Free Reduced Growth Factor Basement Membrane Matrix (Gibco^TM^). hiPSC-NSCs were cultured in 1:1 NeurobasalTM Medium (Gibco^TM^): Advanced Dulbecco’s Modified Eagle Medium: Nutrient Mixture F12 (Advanced DMEM:F12, Gibco^TM^) that was supplemented with Neural Induction supplement (1× end concentration, Gibco^TM^) and 1% penicillin–streptomycin (Gibco^TM^). For neuronal differentiation, cell culture plates were coated with a 20 µg/mL poly-L-ornithine solution (Poly-L-ornithine hydrobromide, Sigma-Aldrich, St. Louis, MO, USA) by incubating for 1 h at 37 °C 5% CO_2_, and this was followed by a Dulbecco’s Phosphate Buffered Saline (DPBS, Gibco^TM^) washing step and further incubating for at least 1 h with a 10 µg/mL laminin solution (Sigma-Aldrich, CC095) at 37 °C 5% CO_2_. Coated plates were stored at 4 °C for up to seven days. Just before seeding of hiPSC-NSCs, laminin was removed, and wells were washed once with DPBS. hiPSC-NSCs were further differentiated into neurons (hiPSC-neurons) using B-27^TM^ Plus Neuronal culture medium (Gibco^TM^) that was supplemented with CultureOne^TM^ Supplement (1× end concentration, Gibco^TM^), 2 mM L-glutamine (Gibco^TM^), 200 µM ascorbic acid (Sigma-Aldrich), 1% penicillin and streptomycin (Gibco^TM^), and 10 ng/mL Recombinant Human Brain-derived Neurotrophic Factor (rhBDNF, Immunotools) and Recombinant human Glial-derived Neurotrophic Factor (rhGDNF, ImmunoTools). Five µM Rock inhibitor (Sigma-Aldrich, Y27632) was added during hiPSC-NSC seeding, i.e., at day 0 of neuronal differentiation, and it was removed between 12 h and 24 h to increase cell survival.

### 2.2. Cell Culture and VZV Propagation

The human retinal pigment epithelial cell line ARPE-19 (ATCC, CRL-2302) was cultured in DMEM:F12 (Gibco^TM^) supplemented with 10% iFBS (Gibco^TM^) and 1% penicillin–streptomycin (Gibco^TM^) and used to propagate the eGFP-ORF23 VZV strain [18]. In this recombinant VZV strain, which is derived from the VZV pOka strain, the minor capsid protein ORF23 is fused to the enhanced green fluorescent protein (eGFP). VZV titers were determined by an infectious foci assay [42]. Briefly, one day beforehand, ARPE-19 cells were seeded at 3 × 10^4^ cells/mL in 96-well plates, after which infected ARPE-19 cells were serially diluted onto the respective uninfected cells. Infectious foci were counted on day 3 (3 dpi) and day 7 post-infection (7 dpi) using an inverted fluorescence microscope (Axio Observer Z.1., Zeiss).

### 2.3. VZV Lysate Preparation

For preparation of VZV lysate, the protocol by Sloutskin et al. [43] was followed with minor modifications. In brief, six T75 cell culture flasks (CELLSTAR^®^, Greiner, Tokyo, Japan) with ARPE-19 monolayers were infected 1:3 with ARPE-19-associated eGFP-ORF23 VZV. When ARPE-19 showed an abundant cytopathic effect, usually at 3 dpi or 4 dpi, the cells were harvested in ice-cold PBS–sucrose–glutamate–serum buffer (8% 10× PBS, 4% (*m/v*) sucrose, 0.08% (*m/v*) glutamate, 8% iFBS) by scraping the cells off the culture flasks. Next, the cells were snap frozen two times in liquid nitrogen and then thawed in a warm water bath at 37 °C. The cell lysate (1.5 mL) was then sonicated for 15 s at 20% amplitude (Vibra-cell^TM^, Sonics & Materials inc.). Finally, the VZV cell lysates were pooled and stored at −80 °C in single-use aliquots. To verify whether ARPE-19 cells were properly lysed, VZV lysate was cultured in DMEM:F12 + 10% iFBS (Gibco^TM^), and this was followed by sevendays incubation at 37 °C 5% CO_2_ during which they were checked for the absence of cell growth. Titers were determined as described above.

### 2.4. Immunofluorescence Staining

hiPSC neurons were fixed with 4% filtered paraformaldehyde (VWR) for 30 min at room temperature (RT), and then washed three times with DPBS (Gibco^TM^). For the staining of intracellular proteins, cells were permeabilized with 0.1% Triton X-100 (Sigma-Aldrich) for 30 min at RT. Subsequently, cells were blocked with serum of the host of the secondary antibody for 1 h, and then washed three times with DPBS. Thereafter, cells were incubated with the primary antibodies for 1 h at RT. Cells were again washed three times with DPBS, and then stained with the secondary antibodies for 1 h at RT. Nuclei were stained with 1 µg/mL 4′, 6-diamidino-2-phenylindole (DAPI, Sigma-Aldrich) for 10 min, and then washed three times with DPBS. Finally, a drop of a 1% solution of 1,4-diazabicyclo [2.2.2]octane (DABCO, Sigma-Aldrich) was added to enhance the lifetime of the dyes. The following antibody combinations for immunofluorescence staining were used: mouse anti-β3-tubulin (2 µg/mL; R&D systems, MAB1195) and rabbit anti-GFAP (1 µg/mL; Abcam, ab7260) in combination with the respective secondary antibodies, goat anti-mouse AF555 (2 µg/mL; Invitrogen, A-21425) and goat anti-rabbit FITC (7.5 µg/mL; Jackson ImmunoResearch, 111-096-045), guinea pig anti-NeuN (1:100; Sigma, ABN90P) in combination with donkey anti-guinea pig Cy3 (7.5 µg/mL; Jackson ImmunoResearch, 706-165-148), chicken anti-peripherin (1:2000; Invitrogen, PA1-1-10012) in combination with goat anti-rabbit FITC (7.5 µg/mL; Jackson ImmunoResearch, 111-096-045), and mouse anti-VZV gE (1:500; Laboratory of Virology and Immunology, GIGA-Infection, Inflammation and Immunity, University of Liège) in combination with rabbit anti-mouse AF555 (4 µg/mL; Invitrogen, A-21427).

### 2.5. Poly(dA:dT) and Poly(I:C) Stimulation and VZV and SeV Inoculation of hiPSC-NSCs and hiPSC Neurons

hiPSC-NSCs and hiPSC neurons, which had been differentiated for 2 to 3 weeks in well plates (CELLSTAR^®^, Greiner), were: (i) transfected with 2 µg/mL poly(dA:dT) (InvivoGen, cat n° tlrl-patn) using Lipofectamine^TM^ 3000 Reagent (Invitrogen, cat n° L3000001) for 24 h or treated with Lipofectamine^TM^ 3000 Reagent alone for 24 h as a control, (ii) stimulated with 10 µg/mL poly(I:C) (InvivoGen, cat n° tlrl-pic) for 24 h without transfection, (iii) stimulated with 10 µg/mL poly(I:C) (InvivoGen, cat n° tlrl-pic) using Lipofectamine^TM^ 3000 Reagent for 24 h or treated with Lipofectamine^TM^ 3000 Reagent alone for 24 h as a control, (iv) inoculated with 1.5 × 10^2^ PFU eGFP-ORF23 VZV lysate or a non-infected cell debris control for either 24 h or 72 h, or (v) inoculated with ARPE-19-associated eGFP-ORF23 VZV at 1:5 ratio or a non-infected ARPE-19 control for 72 h, or (vi) inoculated with Sendai virus (SeV) at MOI 0.1, 1, or 10 for 24 h. For the inoculation of hiPSC-NSCs and hiPSC neurons with SeV, the SeV-GFP4 strain (SKU: S124, ViraTree) was used. In this strain, GFP is expressed from transcription of the GFP gene which is located between the M and F genes of SeV. Cells were harvested using DNA/RNA shield (Zymo Research) and stored at −20 °C for downstream RTqPCR analysis.

### 2.6. Neuronal Differentiation in a Compartmentalized Model, Axonal Infection and IFNα Treatment

Compartmentalized chambers, XonaChips^TM^ (Xona Microfluidics^TM^, XC150), were used to separate the neuronal cell bodies from the axons. Two hundred and twenty-five thousand hiPSC-NSCs were seeded on one side of the XonaChips^TM^ which were coated with poly-L-ornithine and laminin, and the neuronal differentiation was started following the neuronal differentiation protocol. A 10-fold concentration gradient of rhBDNF and rhGDNF was applied to attract the axons through the microgroove barrier (10 µm width, 150 µm length). hiPSC neurons were kept in culture at 37 °C and 5% CO_2_ with the partial medium which was changed every other day containing rhBDNF and rhGDNF to allow extensive axonal outgrowth. At 21 days post-terminal neuronal differentiation (21 dpd), the axon termini of hiPSC neurons were inoculated with 7.3 × 10^3^ PFU ARPE-19-associated eGFP-ORF23 VZV. For the experiments using exogenous IFNα-2a treatment, neuronal cell bodies were mock-treated or pre-treated with 2 × 10^4^ U/mL or 2 × 10^5^ U/mL recombinant human interferon-alpha 2a (rhIFNα-2a, Stem Cell Technologies, Vancouver, BC, Canada) 24 h before the inoculation at 20 dpd. Half-medium changes of the cell body compartment were performed every day for seven days with media containing the vehicle or rhIFNα-2a. Half-medium changes of the axon compartment were performed every other day.

### 2.7. Microscopy and Image Analysis

Immunofluorescent images for characterization of hiPSC neurons were obtained using a PerkinElmer UltraVIEW Vox spinning disk confocal system. ImageJ software (NIH) was used for image processing. Images of hiPSC neurons grown within XonaChips^TM^ were taken using an inverted fluorescence microscope (Axio Observer.Z1, Zeiss, Jena, Germany). For follow-up of VZV spread in neuronal cultures, immunofluorescent images of the cell body and axon compartments of each XonaChip^TM^ were taken using an inverted fluorescence microscope that was equipped with a motorized stage (Axio Observer.Z1 with COLIBRI.2 controller, Zeiss). Each image was composed of 225 tiles that were taken at 20× magnification in brightfield and GFP. Tiles were stitched and exported using ZEN blue software (Zeiss). Prior to analysis, images were cut into two dividing the cell body and axon compartments. ImageJ software (NIH) was used to automatically quantify eGFP+ cells arising from VZV ORF23 expression. Similarly, for quantification of the number of VZV plaques and plaque sizes in ARPE-19 cells, images were taken in brightfield and eGFP+ using an inverted fluorescence microscope (Axio Observer.Z1 with COLIBRI.2 controller, Zeiss, Jena, Germany) and analyzed using ImageJ software (NIH).

### 2.8. Preparation of Neuronal Cell Body Lysates

To prepare cell lysates of hiPSC neurons grown within the XonaChips^TM^, the media were first removed from both of the compartments. Next, a 1% agarose solution in DMEM:F12 was pushed through the axon compartment only, and it was left to solidify at RT. Neuronal cell bodies were harvested using 100 µL DNA/RNA shield (Zymo Research) by pushing the liquid through the cell body compartment and aspirating it. During this process, cultures were observed under an inverted light microscope (Axiovert 40C, Zeiss, Jena, Germany), ensuring total lysis and maximum yield. Cell lysates were stored at −20 °C for downstream applications.

### 2.9. gDNA and RNA Extraction and cDNA Synthesis

Neuronal lysates that were stored in DNA/RNA shield (Zymo Research) were thawed at RT and resuspended in an equal volume of lysis buffer (Zymo Research) just before starting the Quick-DNA/RNA miniprep (Zymo Research) protocol following the manufacturer instructions. Trace gDNA was removed from RNA samples using Monarch^®^ gDNA removal columns (Bioké), and this was followed by a treatment with TURBO^TM^ DNase (Thermo Fisher Scientific, Waltham, MA, USA). Purity of gDNA and RNA samples was checked using a NanoDrop^TM^ 2000 spectrophotometer (Thermo Scientific). RNA samples were immediately used for preparation of cDNA using the SuperScript^TM^ IV First-Strand Synthesis System (Invitrogen^TM^, 18091200) with oligo(dT)_20_ primers (Invitrogen^TM^).

### 2.10. qPCR and RTqPCR

Primers and probes for VZV ORF23 and VZV ORF63 were designed using SnapGene (Insightful Science, LLC, San Diego, CA, USA) and Primer3 software [44] (Table 1). Oli2go design tool [45] was used to select primers and probes for multiplexing. All of the primers were checked for specificity and the optimal annealing, and extension temperature was determined by melting curve analysis in a SYBR Green assay. For the VZV multiplex, a mastermix was prepared consisting of 2x SensiFAST^TM^ Probe No-ROX kit (Bioline, BIO-86005), 500 nM of ORF23, ORF62, ORF63, and hGAPDH (human glyceraldehyde 3-phosphate dehydrogenase) forward (F) and reverse (R) primers, and 100 nM OR23 probe and 250 nM ORF62, ORF63, and hGAPDH probes (Table 1). VLT3-4 was run in singleplex using 800 nM forward and reverse primer and 250 nM probe (Table 1).

For detection of ISGs pre-optimized TaqMan^TM^ assays were used, and the concentrations of them were optimized for multiplexing: (i) duplex consisting of 1 µL of the CXCL10 assay (FAM-MGB, Assay_ID: Hs01124251_g1) and 0.7 µL of the EIF2AK2 assay (VIC-MGB, Assay_ID: Hs00169345_m1), (ii) duplex consisting of 1 µL of the Mx1 assay (FAM-MGB, Assay_ID: Hs00895608_m1), 500 nM hGAPDH forward and reverse primer and 250 nM hGAPDH probe, (iii) triplex consisting of 1 µL of the OAS1 assay (FAM-MGB, Assay_ID: Hs00973635_m1), 0.7 µL of the ISG15 assay (VIC-MGB, Assay_ID: Hs01921425_s1), 500 nM hGAPDH forward and reverse primer, and 250 nM hGAPDH probe, (iv) singleplex consisting of 1 µL of the IFNα-2 assay (FAM-MGB, Assay_ID: Hs00265051_s1). Assays from the TaqMan^TM^ Array Human Interferon Pathway (catalog 4414154) were selected to screen for the expression of human interferons (Table 2).

For RTqPCR reactions 0.1 mg/mL BSA (Thermo Scientific, Waltham, MA, USA) was added to the mastermix. Cycling conditions for all of the qPCR reactions were: 1 cycle of 5 min at 95 °C (hot start, polymerase activation), which was followed by 40 cycles of 10 s at 95 °C (denaturation) and 50 s at 61 °C (annealing and extension). Cycling conditions for all of the RTqPCR reactions were: 1 cycle of 2 min at 95 °C (hot start, polymerase activation), which was followed by 40 cycles of 5 s at 95 °C (denaturation) and 20 s at 61 °C (annealing and extension). All of the PCR reactions were run using a LightCycler^®^ 96 or 480 System (Roche Diagnostics, Basel, Switzerland). For qPCR reactions, no template controls (NTC) were run on each PCR plate. For RTqPCR reactions, no reverse transcriptase controls (NRT) were run for each sample. Relative quantification to hGAPDH was performed using the Pfaffl Method [48], and we accounted for differences in PCR efficiency per target. Non-detects were set at maximum cycling threshold (Ct40) prior to relative quantification calculations being performed. The control group was set as a reference for normalization, i.e., the log2 fold change was determined relative to the mean of the control group. In experiments with no meaningful control group and if the PCR efficiencies of the gene of interest (GOI) and housekeeping gene (HKG) did not differ more than 10%, then ΔCt values are shown (ΔCt = Ct GOI − Ct HKG).

### 2.11. Statistical Analysis

Statistical analysis was performed using JMP^®^ software (SAS Institute). Fluorescent imaging data were root-square transformed, and qPCR and RTqPCR data were log transformed prior to fitting into a Linear Mixed Model, and we accounted for repeated measures, i.e., the combination of independent experiments or repeated measurements for each observation. Statistical outliers (Q1-1.5* IQR; Q3 + 1.5* IQR) were identified for each experiment and excluded from the subsequent statistical analysis. Group means (Least Squares Means Estimates) were compared using Tukey’s Honestly Significant Difference (Tukey HSD) all-pairwise comparison test or Dunnett’s comparisons with control test, and we corrected the data for multiple testing. When only two groups were compared, a paired *t*-test was used. The assumptions for normality and homoscedasticity were met. Corrected *p* values < 0.05 were considered statistically significant

## 3. Results

### 3.1. Differentiation of hiPSC-NSCs into Peripherin-Expressing hiPSC-Neurons

The hiPSC-NSCs were seeded on Geltrex-coated plates before their fixation at day in vitro 2 (DIV 2), or they were seeded on poly-L-ornithine and laminin-coated plates and further differentiated along the neuronal lineage for 3 weeks as described in the Materials and Methods Section. Although some of the hiPSC-NSCs displayed a spontaneous differentiation into β-tubulin III-positive cells (Figure 1A: lower row), a directed differentiation towards the hiPSC neurons resulted in a homogenous population of β-tubulin III-expressing neurons, without contaminating glial fibrillary acidic protein (GFAP)-expressing astrocytes (Figure 1A: upper row). hiPSC-derived astrocytes were used as positive control for the expression of GFAP (data not shown). Additionally, NeuN, which is a marker for mature post-mitotic neurons, displayed nuclear expression in the hiPSC neurons (Figure 1B: upper row). In contrast, the expression of NeuN in the hiPSC-NSCs was low or non-detectable (Figure 1B: lower row). Lastly, the sensory marker peripherin was abundantly expressed in the cell bodies and axons of the hiPSC neurons (Figure 1C: upper row), whilst being absent in the hiPSC-NSCs (Figure 1C: lower row). Concluding, the applied differentiation protocol from the hiPSC-NSCs to the hiPSC-neurons resulted in a population of peripherin-expressing hiPSC neurons resembling sensory neurons in the peripheral nervous system (PNS) that VZV infects in vivo.

### 3.2. Generation of A Compartmentalized Neuronal Culture Model Suitable for Axonal VZV Infection

A compartmentalized model was set up to mimic the in vivo VZV infection route via the infection of the axon termini, which is followed by retrograde axonal transport to the neuronal cell bodies. The hiPSC-NSCs were seeded at one side of poly-L-ornithine and laminin-coated XonaChips^TM^ (Figure 2A), and they were further differentiated into the hiPSC neurons (Figure 2B: right side). The axons were attracted through the microgrooves using a 10-fold growth factor gradient of rhBDNF and rhGDNF (Figure 2B: left side). The immunofluorescence staining procedure for the cell nuclei (DAPI) showed that whilst axons expressing β-tubulin III can migrate through the microgrooves, the cell bodies cannot pass this barrier, thus allowing for axonal VZV infection to occur (Figure 2C). Following 21 days of neuronal differentiation (21 dpd), which allows extensive axonal outgrowth, the axon termini were inoculated with ARPE-19-associated eGFP-ORF23 VZV. This VZV strain expresses a fluorescently labeled capsid protein (ORF23) that allows for the detection of late gene expression. At seven dpi, the immunofluorescence analysis was carried out to detect the nuclei (DAPI) and direct eGFP fluorescence. In addition, a specific staining was performed for VZV gE as this is indicative for an ongoing VZV infection within the neuronal cell body compartment (Figure 2D).

### 3.3. VZV Infection of and Spreading in Neuronal Cultures Do Not Trigger Type I IFN Production

At 21 dpd, the neuronal axon termini were inoculated with cell-associated eGFP-ORF23 VZV which leads to a productive spreading infection in contrast to a cell-free VZV infection at a low MOI which has been shown to induce a quiescent state [35,45]. The VZV-infected cultures displayed a significant time-dependent increase in the number of VZV capsid-associated fluorescent dots in the cell body compartment from three dpi to seven dpi, without them showing a cytopathic effect (Figure 3A,B). This may suggest that the hiPSC neurons do not mount an antiviral response that is sufficient to halt the VZV spread throughout the neuronal cultures. The RTqPCR for VZV ORF23 transcripts at seven dpi confirmed that VZV ORF23 was expressed in these cultures (Figure 3C). Since the VZV infection and its spread throughout the neuronal cell body compartment is asynchronous, it could also be that for some of the hiPSC neurons, it took longer than three days to express a sufficient amount of eGFP for it to be detected by the immunofluorescent imaging. To verify whether the infection that was present in the neuronal cultures was truly productive, the cell body compartment was harvested at seven dpi and it was used for the inoculation of the ARPE-19 cells. The ARPE-19 cells showed an increase in the number of VZV plaques (2/3 experiments) and in the mean plaque sizes (3/3 experiments) from three dpi to five dpi (Figure 3D), thus demonstrating that the hiPSC neurons were indeed productively infected and could transfer the viral particles.

Since type I IFNs are important for early antiviral control, we investigated whether the hiPSC neurons produce type I IFNs upon VZV infection. IFNα-2 is the most well-known IFNα subtype that is also used as a treatment for viral infections. IFNα-2 mRNA was detected in only one out of nineteen neuronal cultures which were infected at their axon termini with ARPE-19-associated VZV and only with a high Ct value (Ct ≥ 35) (data not shown). Besides IFNα-2, the other IFNα subtypes (IFNα-1, -4, -6, -7, -8, -10, -14, -16, -17), IFNβ1, IFNκ, and IFNγ were not expressed either and IFNω was expressed only with a high Ct value (Ct ≥ 35) (Figure 3E). Similarly, the hiPSC-neuronal cultures that were stimulated with poly(dA:dT), a well-known synthetic analogue that activates the IFN-signaling pathway, showed no messenger expression of the IFNα subtypes, except for IFNα-7, although only with a high Ct value (Ct ≥ 35) (Figure 3E). In contrast, the poly(dA:dT) stimulation of the hiPSC neurons did result in the production of low levels of IFNβ1 mRNA (Ct ≥ 30) (Figure 3E). Furthermore, upon direct inoculation of the neuronal cell bodies with ARPE-19-associated eGFP-ORF23 VZV or the eGFP-ORF23 VZV lysate for 72 h, IFNα-2 mRNA was detected in only one out of four cultures that were inoculated with cell-associated VZV, and again, this was with a high Ct value (Ct ≥ 35), and it was not detected upon the infection with the VZV lysate (data not shown). Together, these data show that the hiPSC neurons in our experimental setup do not produce type I IFNs upon VZV infection. Interestingly, IFNAR1 and IFNAR2 were expressed in the hiPSC neurons, indicating that the hiPSC neurons may be able to respond to exogenous interferons (Figure 3E).

### 3.4. IFNα-Production Was Not Detected in hiPSC Neurons

IFNα mRNA was not detected upon the point of the VZV infection or the poly(dA:dT) stimulation of the hiPSC neurons, which may either be related to the specific hiPSC line that was used here or it could rather be a characteristic that is related to the cultured hiPSC neurons. To investigate this, the hiPSC-NSCs, which were used to differentiate into hiPSC neurons, and the hiPSC neurons were both stimulated for 24 h with poly(dA:dT) or poly(I:C), which are both strong inducers of the IFN response. In contrast to the hiPSC neurons, the hiPSC-NSCs showed a significant upregulation of the IFNα-2 transcripts upon their transfection with poly(dA:dT) and poly(I:C) (Figure 4A). This indicates that the lack of IFNα-2 mRNA in the stimulated hiPSC neurons is not the result of a defect in the IFNα production capacity of this specific iPSC-derived cell line.

To investigate whether the lack of IFNα production is solely the result of the characteristics of the hiPSC neurons, or whether this also might be related to the virus-specific properties, the hiPSC-NSCs and the hiPSC neurons were inoculated with SeV, which is known to induce the type I IFN response. Neither the hiPSC-NSCs (except in EXP3, Ct ≥35 and only at MOI 1) nor the hiPSC neurons (except in EXP3 and only at MOI 1) produced IFNα-2 transcripts at 24 hpi (Figure 4B) despite being clearly infected as was evident from GFP expression arising from transcription of the SeV strain (Appendix A). As a control, the hiPSC-NSCs and hiPSC neurons were inoculated with eGFP-ORF23 VZV lysate for 24 h like they were with the SeV strain. Again, no IFNα-2 mRNA transcripts were detected in the hiPSC neurons that were inoculated with VZV nor in the hiPSC-NSCs (Figure 4B and Appendix A).

Taken together, these data suggest that hiPSC neurons are overall poor IFNα producers upon an immunogenic stimulation and upon a viral infection.

### 3.5. VZV Infection Did Not Result in ISG Upregulation in hiPSC Neurons

Since the hiPSC neurons did not produce type I IFNs upon VZV infection, we questioned whether the ISGs downstream of this pathway were not upregulated either or perhaps they may have been upregulated through a non-canonical pathway. Hereto, the hiPSC neurons were inoculated and cultivated for 72 h with ARPE-19-associated eGFP-ORF23 VZV or eGFP-ORF23 VZV lysate and they were compared to the mock-inoculated cultures. The poly(dA:dT) stimulation served as a positive control for the production of the ISGs. Upon the poly(dA:dT) stimulation, the PKR, ISG15, Mx1, OAS1, and CXCL10 mRNAs were indeed significantly upregulated. In contrast, neither the ARPE-19-associated VZV nor the VZV lysate upregulated the messenger expression of any of the ISGs that were tested (Figure 5A). The expression of CXCL10 mRNA in the VZV- and mock-inoculated cultures was below the limit of detection (Ct > 40), and the expression of OAS1 mRNA was either very low (Ct ≥ 35) or it was not detected (Ct > 40). The expression of VZV ORF23 mRNA in the cultures that were inoculated with ARPE-19-associated VZV and VZV lysate indicated a successful infection of the neuronal cultures (Figure 5B). To examine whether the inability of VZV to upregulate the ISGs is related to VZV or rather, whether it is a general feature of the viral infection of the cultured hiPSC neurons, the hiPSC neurons were inoculated with SeV and compared to the VZV-inoculated cultures. The hiPSC neurons were (i) inoculated with an eGFP-ORF23 VZV lysate for 24 h and compared to the mock-inoculated cultures or (ii) inoculated with SeV for 24 h at different MOIs. The PKR, ISG15, Mx1, OAS1, and CXCL10 mRNAs were significantly upregulated in the cultures that were inoculated with SeV. In contrast, the infection with the VZV lysate did not upregulate ISG expression in the hiPSC neurons (Figure 5C). The expression of VZV ORF23 mRNA in the cultures that were inoculated with the VZV lysate (Figure 5D) and the representative images of the hiPSC neurons that were inoculated with the VZV lysate and SeV (Appendix A: hiPSC neurons) indicate successful VZV and SeV infections. In conclusion, we observed that the hiPSC neurons express basal levels of ISG transcripts, which are not upregulated upon a VZV infection but, in contrast, they are upregulated upon poly(dA:dT) stimulation and upon an SeV infection.

### 3.6. Exogenous IFNα Limited VZV Spread in the Neuronal Cell Body Compartment upon Axonal Infection

Considering the fact that IFNα was not produced by hiPSC neurons upon the VZV infection, but that hiPSC-neurons can upregulate the ISGs upon their stimulation, we evaluated the effect of exogenous IFNα treatment on the VZV spread in the neuronal cultures. The neuronal cell body compartment was either mock-treated (untreated control cultures, UTC) or treated with 2 × 10^4^ U/mL (lower dose) or 2 × 10^5^ U/mL (higher dose) IFNα-2a 24 h prior to the inoculation of the neuronal axon termini with ARPE-19-associated eGFP-ORF23 VZV. The cultures were maintained for up to seven dpi with us performing daily medium changes of the cell body compartment containing IFNα-2a or the vehicle (Figure 6A). In the untreated control cultures, the infection was observed at three dpi and the extensive spreading of VZV in the cell body compartment was seen at seven dpi. The lower concentration IFNα-2a had no effect on the initial infection as observed at three dpi, while the higher concentration already reduced the infection at this time. At seven dpi, the effects were more prominent, with the lower dose IFNα-2a resulting in a significant reduction in the eGFP+ dots arising from the VZV ORF23 expression when they were compared to the untreated control cultures and an even stronger reduction in this with the higher dose IFNα-2a (Figure 6B).

We aimed to confirm the immunofluorescence data by the relative quantification of VZV mRNA in the cells from the cell body compartment. When they were compared to untreated control cultures, the relative number of VZV ORF23 transcripts was significantly lower in the cultures that were treated with 2 × 10^5^ U/mL IFNα-2a (Figure 7A). Besides ORF23, the RTqPCR for VZV ORF62, ORF63, and VLT was also performed to provide additional information on the VZV replication and the confirmation of the reduction of the VZV spread. VZV ORF62, ORF63, and VLT all showed a significant and dose-dependent decrease in the number of transcripts upon the IFNα-2a treatment (Figure 7A). We also examined whether the VZV transcript expression was directly correlated with the level of VZV replication and virus spreading through the quantification of the gDNA. Significantly fewer VZV ORF23, ORF62, and ORF63 gDNA copies were detected in the cultures that were treated with IFNα-2a when they were compared to the untreated controls (Figure 7B).

In summary, these data show that although the hiPSC neurons do not produce IFNα upon VZV infection, the exogenous IFNα-2a, which was administered before and during the infection, effectively reduces the infection and controls the VZV spread in the neuronal cultures.

### 3.7. ISGs Were Upregulated in VZV-Infected hiPSC Neurons Treated with IFNα

The reduction in the VZV spread and the levels of VZV gDNA and VZV mRNA upon the IFNα-2a treatment suggest that an IFNα-mediated response that is able to control the VZV infection is upregulated. Thus, we examined the ISG expression in the VZV-infected neuronal cultures which were both untreated and treated with IFNα-2a. The PKR, OAS1, ISG15, and Mx1 mRNAs were detected in the untreated control cultures, yet they were significantly upregulated upon the IFNα-2a treatment: the expression of PKR mRNA was increased by approximately 2 log folds, whereas for the OAS1, ISG15, and Mx1 mRNAs, a 6 to 10 log fold increase was observed (Figure 8). Furthermore, we also determined mRNA transcripts for one ISG that is involved in attracting T cells to the site of the infection (CXCL10) to evaluate whether there was an upregulation of the transcripts without there being direct antiviral effects. The CXCL10 expression in the untreated control cultures was either very low (Ct ≥ 35) or it was not detectable (Ct > 40), and the treatment with IFNα-2a resulted in a consistent increase in this (Ct < 35) (Figure 8). Thus, the exogenous IFNα treatment of the hiPSC neurons effectively upregulates the production of the ISGs.

## 4. Discussion

Studying VZV–host interactions in *bona fide* human neuronal models that were designed to resemble the in vivo situation as close as possible remains a major challenge. Previous models, such as those described in Markus et al. [38], illustrate the use of compartmentalized chambers in combination with embryonic stem cell-derived neurons. In this study, we developed a model that was based on iPSC-derived neurons, which also allowed the physiologically relevant axonal infection of the neurons to occur. In addition, this model also has the potential to be used to study in more detail the virus–host interactions in more complex models, e.g., by allowing for a co-culture with autologous immune cells and even with patient-derived cells.

Following the VZV inoculation of the hiPSC neurons at their axon termini, the VZV spread throughout the neuronal cell body compartment was reflected in an increase in eGFP expression, arising from VZV ORF23 expression, over time. Thus, hiPSC neurons cannot limit the VZV spread on their own. Interestingly, type I IFNs were not detected in these VZV-infected neuronal cultures. Furthermore, the direct inoculation of the neurons with cell-associated VZV or a VZV lysate did not result in the production of IFNα-2 mRNA either. It was previously described that VZV-infected skin cells also do not seem to produce IFN, as opposed to adjacent uninfected skin cells [2]. Surprisingly, neither the poly(dA:dT), the poly(I:C) stimulation, nor the SeV infection of the hiPSC neurons induced IFNα-2 mRNA production. However, the poly(dA:dT) stimulation resulted in the production of low levels of IFNβ mRNA. Together, this suggests that hiPSC neurons are unable to express IFNα at detectable levels. The resistance of hiPSC-neurons to produce high levels of type I IFNs, and in particular IFNα, could reflect an intrinsic protection mechanism to avoid neuronal cell death. Indeed, IFNα treatment in the context of multiple sclerosis and chronic myeloid leukemia has been associated with neurotoxicity, which is in contrast to IFNβ that is used for the treatment of multiple sclerosis [49,50]. Moreover, another study investigating type I IFN production upon the infection of the mouse central nervous system with the neurotropic viruses Theiler’s virus and La Crosse virus showed that IFNβ and some IFNα subtypes were produced by the neurons, but only in a very small subset of around 3% of the infected neurons [51].

Besides the lack of type I IFN production upon VZV infection, we also observed a lack of ISG upregulation. This could be expected since the ISGs are located downstream in the canonical type I IFN pathway. To further investigate whether the lack of ISG upregulation was related to the VZV infection or whether it is a characteristic of the hiPSC neurons, the hiPSC neurons were stimulated with strong synthetic ISG inducers: poly(dA:dT) and poly(I:C). This synthetic stimulation did induce ISG upregulation, suggesting that the hiPSC neurons can produce ISGs. The hiPSC neurons were also inoculated with SeV, a virus that is known to be a strong inducer of type I IFNs and ISGs. In contrast to the VZV infection, the SeV infection did upregulate the production of the ISGs. However, it should be noted that the GFP signal arising from the SeV infection was much higher than that of the VZV infection. Thus, we cannot exclude that the high MOI of SeV resulted in the stimulation of some of the ISGs, while the VZV infection did not do this.

Although none of the stimuli that were tested, i.e., the stimulation with poly(dA:dT) or poly(I:C) and the SeV infection, led to the production of IFNα, they resulted in the upregulation of the ISGs, thus, we conclude that the cultured hiPSC neurons are poor IFNα producers, yet they are capable of producing ISGs. Since we observed that IFNβ was induced upon the poly(dA:dT) stimulation, one possibility is that the ISG-induction mechanism in the hiPSC neurons may occur primarily via IFNβ and not via IFNα production.

The finding that the ISGs are upregulated upon the SeV infection and upon the poly(dA:dT) and poly(I:C) stimulation, yet they are not upregulated upon the VZV infection, may be the result of a VZV immune evasion mechanism. Interestingly, Vandevenne et al. showed that the serine threonine kinase that is encoded by the ORF47 gene induces an atypical phosphorylation of IRF3, inhibiting the self-dimerization of IRF3 that is required for efficient IFNβ induction [52]. Hence, in our hiPSC neurons, a similar VZV-evasive mechanism could be at play. Several VZV evasion mechanisms affecting the IFN–ISG axis have already been described in non-neuronal cells [52,53,54,55,56], however, this is not the case for neuronal cells. It merits further investigation to examine to what extent the VZV’s innate evasion mechanisms are related to the cell type and whether they are host-dependent. For VZV to persist in humans, a delicate balance between immune evasion and recognition is required. The substantial elimination of the VZV-infected neurons and thus, of the VZV reservoir is not desirable, neither for the virus nor for the host.

We observed that upon VZV infection, the hiPSC neurons did not mount an IFN-mediated response and could not control the virus’ replication. The exogenous IFNα-2a treatment, however, was very potent in reducing the VZV infection and its spread throughout the neuronal cultures as evidenced by fluorescent imaging and the VZV gDNA and mRNA levels. Interestingly, we did detect the expression of IFNAR1 and IFNAR2 in the hiPSC neurons, which already suggested that the neurons can respond to type I IFNs. In a previous report by Como et al. in which the hiPSC neurons were pre-treated with IFNα prior to their inoculation with cell-free vaccine-strain VZV, the VZV spread was also reduced [57]. However, in that report, the neuronal cell bodies were directly inoculated with cell-free VZV, which is in contrast to the setup in our experiment that more closely mimics the in vivo situation through the cell-associated axonal VZV infection. Nonetheless, it should be noted that the infection of the neurons could also (primarily) occur via infected T cells [5,6]. Finally, our model can be further optimized by replacing the human retinal pigment epithelial cell line (ARPE-19) that was used for the VZV inoculations of the axon termini with iPSC-derived keratinocytes or epithelial cells, which could then produce type I IFNs upon their VZV infection as it would also occur in vivo.

It was previously shown that an axonal infection with cell-associated VZV leads to a productive spreading infection, whereas an axonal infection with cell-free VZV at a low MOI can induce a quiescent state [37,58]. Similarly, in our setup in which the axon termini were inoculated with cell-associated VZV, a productive infection was observed as shown by the eGFP+ expression arising from VZV ORF23 transcription and by the detection of VZV mRNAs. Interestingly, De Regge et al. reported that IFNα can induce a quiescent herpes simplex virus-1 (HSV-1) and pseudorabies virus (PRV) infection in porcine TG upon their axonal infection [59]. In contrast, in our experiments where we combined a cell-associated VZV infection with an IFNα treatment, productive infections of the hiPSC neurons were observed. Upon the IFNα treatment, the expressions of the VZV transcripts ORF23, ORF62, and ORF63 were reduced, yet they did still increase from three dpi to seven dpi (data not shown). In addition, VLT, which is transcribed during latency in human TG, but that is also expressed during lytic infection [24], was detected both in the untreated and IFNα-treated cultures. Possibly, higher concentrations and/or different interferons together with a relatively low MOI of highly purified cell-free VZV or cell-associated VZV may be needed to induce a quiescent VZV infection. Indeed, the inoculation of the axon termini with cell-associated VZV resulted in a productive VZV infection in the cell body compartment, and thus represented an artificial state. Furthermore, a quiescent state may also be induced through the direct delivery of VZV particles to the neuronal cell bodies by infected T cells. This infection route was not explored in this study.

Since the IFNα treatment reduced the VZV spread in the hiPSC-neuronal cultures, we hypothesized that ISGs were upregulated. Indeed, PKR, ISG15, OAS, Mx1, and CXCL10 transcripts were all significantly upregulated. It might be tempting to hypothesize that the upregulation of the ISGs is (one of) the mechanism(s) leading to the limitation of the VZV spread in the cultures that were treated with IFNα, but this requires more research. Little is known about the antiviral effector functions of ISGs in the context of alpha-herpesviruses. One report shows that mice that are deficient in ISG15 have an increased susceptibility to infection with several viruses including HSV-1 [60]. Another report describes that while the expression of PKR and its phosphorylation level is not modulated strongly during ongoing VZV replication, it is increased in the presence of IFNβ, which is concurrent with the limitation of VZV replication [61]. Furthermore, Mx1 reduces the HSV-1 replication in human fibroblasts [62], and CXCL10-null mice that are infected with HSV-1 and -2 have reduced numbers of NK and CD8+ T cells at the site of the infection and increased rates of viral replication [63,64].

## 5. Conclusions

In summary, we showed that the exogenous administration of IFNα to human iPSC-derived neurons in compartmentalized chambers effectively limits the VZV spread in these cultures and that ISGs are significantly upregulated. This shows that neurons adequately respond to the INFα treatment, thereby underscoring its importance in early innate immune control. However, we observed that although the hiPSC neurons upregulated the ISG transcription upon their stimulation with synthetic ISG inducers and upon the SeV infection, they do not produce detectable levels of IFNα mRNA. Thus, we suggest that while the cultured hiPSC neurons are good IFNα responders, they are poor IFNα producers. This may imply that other cell types within the PNS are essential as IFNα producers. Macrophages or dendritic cells are perhaps the most evident cell types in the PNS that could produce IFNα upon a VZV infection or reactivation. However, satellite glial cells (SGCs) could also produce IFNα in response to a VZV infection. Indeed, Steain et al. observed MHC I upregulation, but more significantly MHC II upregulation on SGCs within both the reactivated and neighboring ganglia from patients who were suffering from active HZ at the time of their death [65]. This indicates that SGCs can function as non-professional antigen-presenting cells in the PNS. In addition, Arnold et al. reported the upregulation of OAS1, ISG15, and Mx1 in the sensory ganglia that were infected with SVV, and they argue this innate immune response is mediated by the SGCs [66]. Future experiments whereby hiPSC-derived immune cells (macrophages, dendritic cells, and/or SGCs) are added to the neuronal cultures could elucidate which cell type(s) can adequately respond to a VZV infection by the production of IFNα.

## Figures and Tables

**Figure 1 viruses-14-02517-f001:**
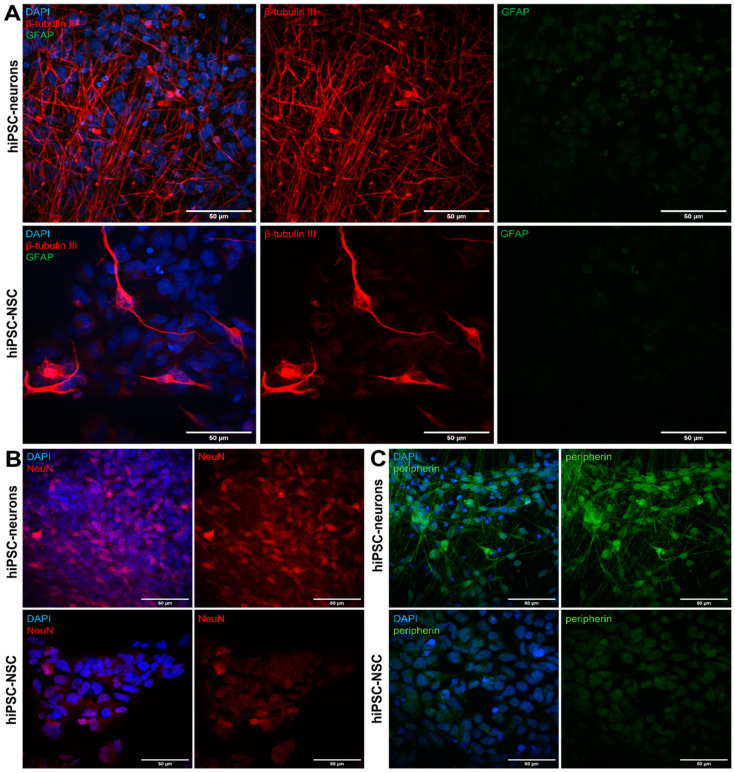
Differentiation of hiPSC-NSCs into peripherin-expressing hiPSC neurons. (**A**) hiPSC-NSCs and hiPSC neurons were stained with mouse anti-β-tubulin III and rabbit anti-GFAP primary antibodies in combination with goat anti-mouse AF555 (red) and goat-anti rabbit FITC (green) secondary antibodies. Nuclei were stained with DAPI (blue). Representative images from three independent repeats of hiPSC-NSCs at DIV 2 and hiPSC neurons at 21 dpd. (**B**) hiPSC-NSCs and hiPSC neurons were stained with guinea pig anti-NeuN and secondary donkey anti-guinea pig Cy3 (red) antibody. Nuclei were stained with DAPI (blue). Representative images from three independent repeats of hiPSC-NSCs at DIV 2 and hiPSC neurons at 21 dpd. (**C**) hiPSC-NSCs and hiPSC neurons were stained with chicken anti-peripherin and secondary goat anti-rabbit FITC (green) antibody. Nuclei were stained with DAPI (blue). To verify specificity of the neuronal markers, secondary antibody controls were prepared for hiPSC-NSCs and hiPSC neurons (data not shown). Representative images from three independent repeats of hiPSC-NSCs at DIV 2 and hiPSC neurons at 21 dpd.

**Figure 2 viruses-14-02517-f002:**
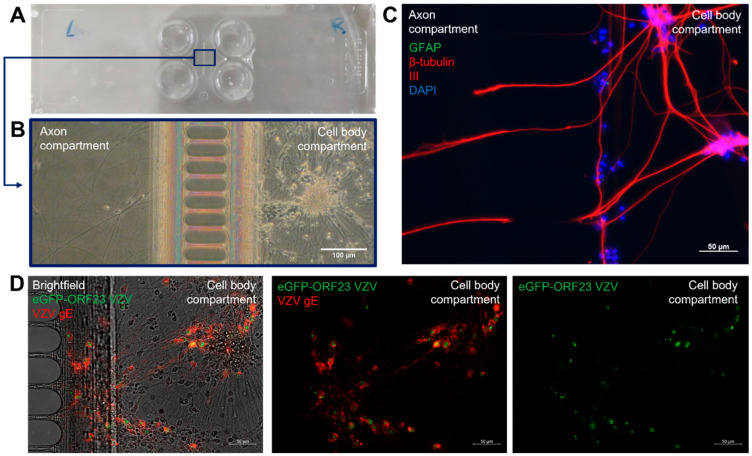
Generation of a compartmentalized neuronal culture model suitable for axonal VZV infection. (**A**) Image of a XonaChipTM (XC150). The two left wells and two right wells are separated from each other through a microgroove barrier (150 µm length, 10 µm width). (**B**) Brightfield image of a hiPSC-derived neuronal culture showing separation of cell bodies from axon termini. Representative image (at 18 dpd) with images that were acquired between 7 dpd and 21 dpd. (**C**) hiPSC-neuronal cultures were stained with mouse anti-β-tubulin III and rabbit anti-GFAP primary antibodies in combination with goat anti-mouse AF555 (red) and goat-anti rabbit FITC (green) secondary antibodies. Nuclei were stained with DAPI (blue). Representative image (at 13 dpd) with images that were acquired between 7 dpd and 21 dpd. (**D**) Spreading of eGFP-ORF23 VZV in the cell body compartment following inoculation of axon termini. Images show expression of eGFP, arising from the VZV capsid protein ORF23, and VZV gE which was stained with mouse anti-VZV gE primary antibody in combination with rabbit anti-mouse AF555 (red) secondary antibody. Representative image at 7 dpi (28 dpd).

**Figure 3 viruses-14-02517-f003:**
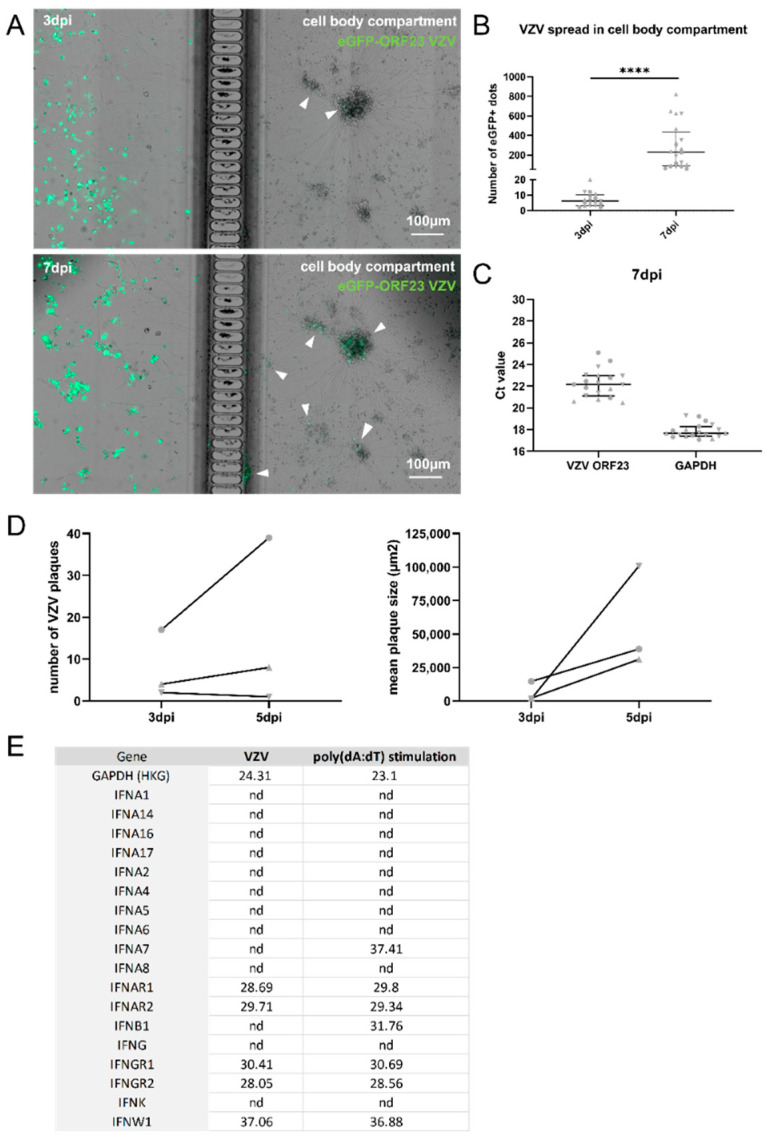
VZV infection of and spreading in neuronal cultures does not trigger type I IFN production. (**A**) Representative images of sections of neuronal cultures at 3 dpi (upper panel) and 7 dpi (lower panel) inoculated at their axon termini with 7.3 × 10^3^ PFU ARPE-19-associated eGFP-ORF23 VZV. Arrow heads indicate eGFP+ dots in the neuronal cell body compartment arising from VZV ORF23 expression. (**B**) Quantification of the number of eGFP+ dots arising from VZV ORF23 expression in the cell body compartment at 3 dpi and 7 dpi. Median number of eGFP+ dots across experiments at 3 dpi: 6 and at 7 dpi: 231. Each symbol represents the number of eGFP+ dots in the cell body compartment of a single culture originating from one of three independent biological repeats (3 dpi: Experiment 1: circles, n = 6; Experiment 2: triangles, n = 7, Experiment 3: upside down triangles, n = 5; 7 dpi: Experiment 1: circles, n = 6, Experiment 2: triangles n = 8, Experiment 3: upside down triangles, n = 5). Lines with whiskers indicate the median with interquartile ranges from the three independent experiments combined. The *p* values reported were obtained from a Linear Mixed Model that was adjusted for multiple testing using Dunnett’s comparisons with control test. (**C**) Ct values for VZV ORF23 and hGAPDH mRNA transcripts. Each symbol represents the Ct value of a single culture originating from one of three independent biological repeats (Experiment 1: circles, n = 7, Experiment 2: triangles n = 7, Experiment 3: upside down triangles, n = 5). Lines with whiskers indicate the median with interquartile ranges from the three independent experiments combined. (**D**) Quantification of the number of VZV plaques (**left**) and mean plaque size (**right**) upon inoculation of ARPE-19 cells with VZV-infected neuronal cell bodies at 3 dpi and 5 dpi. Data are obtained from three independent repeats (Experiment 1: circles, n = 1, Experiment 2: triangles n = 1, Experiment 3: upside down triangles, n = 1). (**E**) Ct values for type I and II IFNs and receptors are shown. Data are obtained from a pooled sample originating from three independent biological repeats. **** *p* < 0.0001, nd: not detected.

**Figure 4 viruses-14-02517-f004:**
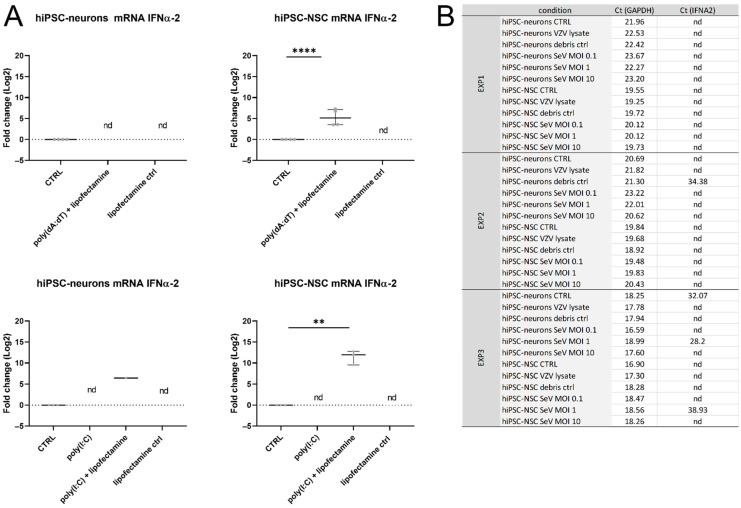
IFNα production is not detected in hiPSC neurons. (**A**) hiPSC-neurons do not produce detectable levels of IFNα-2 mRNA upon poly(dA:dT) or poly(I:C) stimulation. In contrast, in hiPSC-NSCs, IFNα-2 mRNA is significantly upregulated upon stimulation with poly(dA:dT) and poly(I:C) via transfection. Cell lysates were harvested at 24 h post stimulation and analyzed by RTqPCR. Transcript expression (log2) was set out relative to the mean of the untreated non-infected control group (CTRL), i.e., fold change. Each dot represents the log2 fold change of a single culture originating from one of four (poly(dA:dT) stimulation) or three (poly(I:C) stimulation) independent biological repeats. Lines with whiskers indicate the median with interquartile range. The *p* values were obtained from a paired *t*-test. (**B**) Raw Ct values for GAPDH (HKG) and IFNα-2 mRNA transcripts from each experiment are shown in the table. ** *p* < 0.01, **** *p* < 0.0001, nd: not detected.

**Figure 5 viruses-14-02517-f005:**
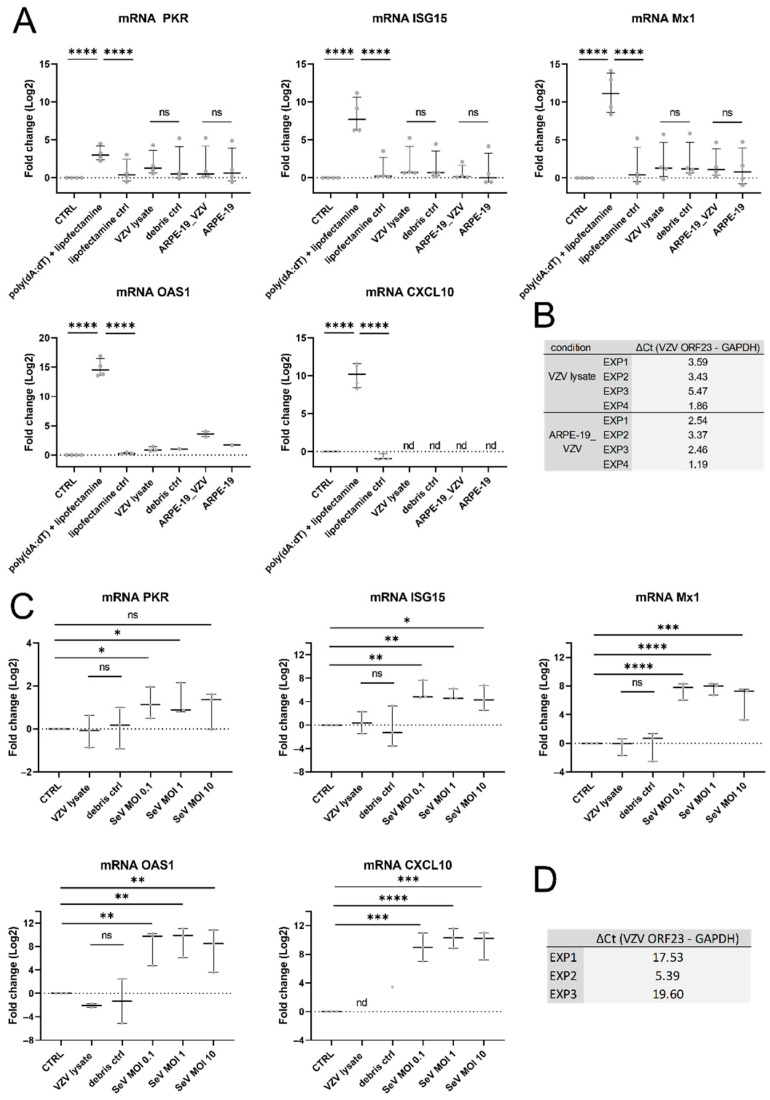
VZV infection does not result in ISG-upregulation in hiPSC neurons. (**A**,**B**) hiPSC neurons were inoculated with ARPE-19-associated eGFP-ORF23 VZV (ARPE-19_VZV) at a 1:5 ratio or with 1.5 × 10^2^ PFU eGFP-ORF23 VZV lysate or their respective control for 72 h. Poly(dA:dT) stimulation for 24 h served as a positive control for the induction of ISGs. Cell lysates were analyzed by RTqPCR. (**A**) Transcript expression (log2) was set out relative to the mean of the untreated non-infected control group (CTRL), i.e., fold change. Each dot represents the log2 fold change of a single culture originating from one of four independent biological repeats. Lines with whiskers indicate the median with interquartile range. The *p* values reported were obtained from a Linear Mixed Model that was adjusted for multiple testing using Dunnett’s comparisons with control test. (**B**) ΔCt values for VZV ORF23 mRNA (Ct VZV ORF23-Ct hGAPDH) are shown. (**C**,**D**) hiPSC neurons were (i) inoculated with 1.5 × 10^2^ PFU eGFP-ORF23 VZV lysate or with the debris control for 24 h, or (ii) inoculated with SeV-GFP4 (SeV) for 24h at different MOIs. Cell lysates were analyzed by RTqPCR. (**C**) Transcript expression (log2) was set out relative to the mean of the untreated non-infected control group (CTRL), i.e., fold change. Each dot represents the log2 fold change of a single culture originating from one of three independent biological repeats. Lines with whiskers indicate the median with interquartile range. The *p* values reported were obtained from a Linear Mixed Model that was adjusted for multiple testing using Tukey HSD all-pairwise comparison test. (**D**) ΔCt values for VZV ORF23 mRNA (Ct VZV ORF23-Ct hGAPDH) are shown. * *p* < 0.05, ** *p* < 0.01, *** *p* < 0.001, **** *p* < 0.0001, ns: not significant, nd: not detected (originating from Ct > 40 for the GOI).

**Figure 6 viruses-14-02517-f006:**
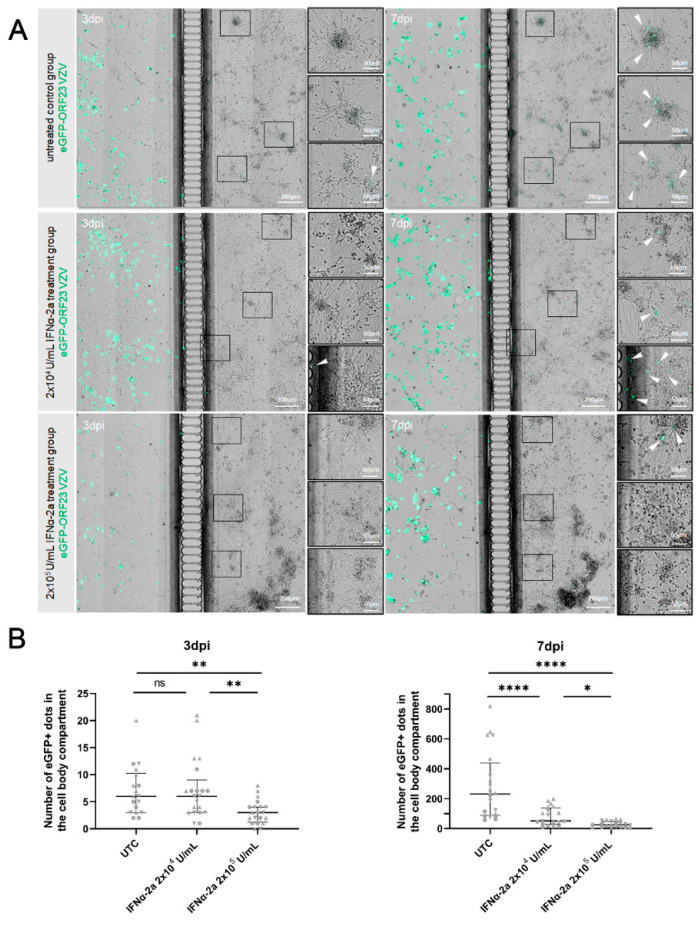
Exogenous IFNα limits VZV spread in the neuronal cell body compartment upon axonal infection. (**A**) Representative images of sections of neuronal cultures at 3 dpi (left panels) and 7 dpi (right panels) are shown for each condition: untreated control cultures (UTC: upper panel), 2 × 10^4^ U/mL (lower dose: middle panel) and 2 × 10^5^ U/mL (higher dose: lower panel) IFNα-2a treatment group. Axon termini were inoculated with 7.3 × 10^3^ PFU ARPE-19-associated eGFP-ORF23 VZV. Arrow heads indicate eGFP+ dots in the neuronal cell body compartment arising from VZV ORF23 expression. (**B**) Quantification of the number of eGFP+ dots arising from VZV ORF23 expression in the cell body compartment. Median number of eGFP+ dots across experiments at 3 dpi: UTC: 6, 2 × 10^4^ U/mL IFNα-2a treatment group: 6, 2 × 10^5^ U/mL IFNα-2a treatment group: 3. Median number of eGFP+ dots across experiments at 7 dpi: UTC: 231, 2 × 10^4^ U/mL IFNα-2a treatment group: 51.5, 2 × 10^5^ U/mL IFNα-2a treatment group: 25. Each symbol represents the number of eGFP+ dots in the cell body compartment of a single culture originating from one of three independent biological repeats (3 dpi: Experiment 1: circles, n = 6/7/6; Experiment 2: triangles, n = 7/8/8; Experiment 3: upside down triangles, n = 5/6/6; 7 dpi: Experiment 1: circles, n = 6/6/6; Experiment 2: triangles, n = 8/8/8; Experiment 3: upside down triangles, n = 6/6/6). Lines with whiskers indicate the median with interquartile range from the three independent experiments combined. The *p* values reported were obtained from a Linear Mixed Model that was adjusted for multiple testing using Tukey HSD all-pairwise comparison test. * *p* < 0.05, ** *p* < 0.01, **** *p* < 0.0001. ns: not significant.

**Figure 7 viruses-14-02517-f007:**
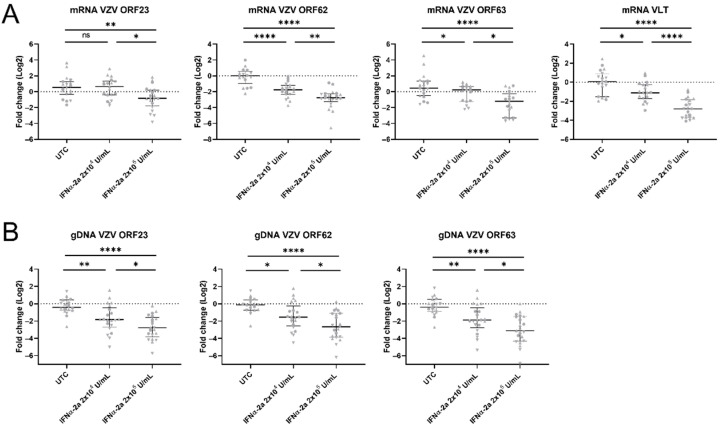
IFNα reduces VZV transcript expression and VZV genome replication. (**A**) Relative quantification of mRNA transcripts in the cell body compartment at 7 dpi by RTqPCR. Transcript expression (log2) was set out relative to the mean of the untreated control group (UTC), i.e., fold change. Each symbol represents the log2 fold change of a single culture originating from one of three independent biological repeats (Experiment 1: circles, ORF23 n = 7/7/7, ORF62 n = 7/7/7, ORF63 n = 7/7/7, VLT n = 7/7/7; Experiment 2: triangles, ORF23 n = 7/8/7, ORF62 n = 7/8/8, ORF63 n = 7/6/7, VLT n = 7/8/7; Experiment 3: upside down triangles, ORF23 n = 5/4/6 ORF62 n = 3/5/5, ORF63 n = 5/3/5, VLT n = 5/4/6). Lines with whiskers indicate the median with interquartile range from the three independent experiments combined. The *p* values reported were obtained from a Linear Mixed Model that was adjusted for multiple testing using Tukey HSD all-pairwise comparison test. (**B**) Relative quantification of gDNA copies in the cell body compartment at 7 dpi by qPCR. The number of gDNA copies (log2) was set out relative to the mean of the untreated control group (UTC), i.e., fold change. Each symbol represents the log2 fold change of a single culture originating from one of three independent biological repeats (Experiment 1: circles, ORF23 n = 6/7/7, ORF62 n = 5/7/7, ORF63 n = 6/7/7; Experiment 2: triangles, ORF23 n = 7/8/8, ORF62 n= 7/8/8, ORF63 n = 7/8/8; Experiment 3: upside down triangles, ORF23 n = 6/6/6, ORF62 n = 6/6/6, ORF63 n = 6/6/6). Lines with whiskers indicate the median with interquartile range from the three independent experiments combined. The *p* values reported were obtained from a Linear Mixed Model that was adjusted for multiple testing using Tukey HSD all-pairwise comparison test. * *p* < 0.05, ** *p* < 0.01, **** *p* < 0.0001. ns: not significant.

**Figure 8 viruses-14-02517-f008:**
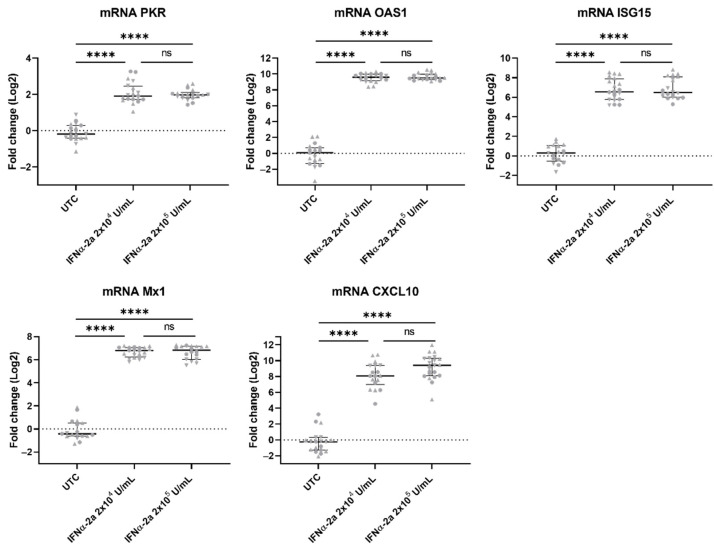
ISGs are upregulated in VZV-infected hiPSC neurons that are treated with IFNα. Neuronal cell body lysates were harvested at 7 dpi and analyzed by RTqPCR. Transcript expression (log2) was set out relative to the mean of the untreated control group (UTC), i.e., fold change. Each symbol represents the log2 fold change of a single culture originating from one of three independent biological repeats (Experiment 1: circles, PKR n = 7/7/6, OAS1 n = 7/7/7, ISG15 n = 6/7/7, Mx1 n = 7/7/7, CXCL10 n = 7/7/7; Experiment 2: triangles, PKR n = 7/8/7, OAS1 n = 8/8/8, ISG15 n = 7/8/7, Mx1 n = 8/8/6, CXCL10 n = 7/8/8; Experiment 3: upside down triangles, PKR n = 5/5/5, OAS1 n = 4/5/5, ISG15 n = 4/5/6, Mx1 n = 4/5/5, CXCL10 n = 5/3/6). Lines with whiskers indicate the median with interquartile range from the three independent experiments combined. The *p* values reported were obtained from a Linear Mixed Model that was adjusted for multiple testing using Tukey HSD all-pairwise comparison test. **** *p* < 0.0001. ns: not significant.

**Table 1 viruses-14-02517-t001:** Primer and probe sequences that were used in VZV multiplex.

Gene	Primer/Probe Sequence	Reference
Human GAPDH_F	5’-CACATGGCCTCCAAGGAGTAA-3’	[46]
Human GAPDH_R	5’-TGAGGGTCTCTCTCTTCCTCTTGT-3’	
Human GAPDH_Probe	5’-(Cy5)CTGGACCACCAGCCCCAGCAAG(IAbRQSp)-3′	
VZV ORF23_F	5′-CTTCTGGACAACAACCGCAA-3′	n/a
VZV ORF23_R	5′-CAGATTGTCCCGTGTGTGAC-3′	
VZV ORF23_Probe	5′-(TexRed-XN)-ACTGTCCAGCCAACAACCGG-(IabRQSp)-3′	
VZV ORF62_F	5′-CCTTGGAAACCACATGATCGT-3′	[47]
VZV ORF62_R	5′-AGCAGAAGCCTCCTCGACAA-3′	
VZV ORF62_Probe	5′-(HEX)-TGCAACCCGGGCGTCCG-(ZEN/IabRQSp)-3′	
VZV ORF63_F	5′-GATGGTGGTGAAGACGA-3′	n/a
VZV ORF63_R	5′-AATCGGTGCTCTCCTCT-3′	
VZV ORF63_Probe	5′-(FAM)-CGGAATCATCGGACGGGGAAG-(ZEN/IabRQSp)-3′	
VLT 3-4_F	5′-AATCGAGCCATACACCACCG-3′	[24]
VLT 3-4_R	5′-TGTATTCGGGCATGGACCTC-3′	
VLT 3-4_Probe	5′-(FAM)-GATCGAACAGCAGATGGATTGCA-(ZEN/IabRQSp)-3′	

**Table 2 viruses-14-02517-t002:** Human interferon pathway: Assay IDs and targets.

Assay ID	Target
Hs00855471_g1	IFNA1
Hs00533748_s1	IFNA14
Hs03005057_sH	IFNA16
Hs00819693_sH	IFNA17
Hs02621172_s1	IFNA2
Hs01681284_sH	IFNA4
Hs00818220_s1	IFNA5
Hs00819627_s1	IFNA6
Hs01652729_s1	IFNA7
Hs00932530_s1	IFNA8
Hs01066116_m1	IFNAR1
Hs00174198_m1	IFNAR2
Hs00277188_s1	IFNB1
Hs00989291_m1	IFNG
Hs00166223_m1	IFNGR1
Hs00194264_m1	IFNGR2
Hs00737883_m1	IFNK
Hs00357857_s1	IFNW1

## Data Availability

Not applicable.

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
