# Peer review of "Activation of Interferon-Stimulated Genes following Varicella-Zoster Virus Infection in a Human iPSC-Derived Neuronal In Vitro Model Depends on Exogenous Interferon-α"

_viruses, 2022, doi:10.3390/v14112517_

Round 1

Reviewer 1 Report

The paper by Boeren et al uses human iPSC-derived neurons to study the role of type I interferons in limiting the spread of VZV in the nervous system.  The manuscript is an elegant one, that is well-written and superbly discussed.  The authors utilize a compartmented system that enables hiPSC-derived neurons to be infected via their axons, which the authors presume mimics the in vivo situation.  Surprisingly, the authors discover that hiPSC-derived neurons are poor producers of INFa, but are nevertheless good responders to INFa, when that molecule is applied to them exogenously.  That is, the hiPSC-derived neurons are very capable of expressing interferon-stimulated genes.  The authors conclude that as INFa applied from an exogenous source, such as glia, satellite cells, or macrophages is necessary to retard the spread of VZV in vivo.

There are a small number of issues with which the authors have not adequately come to grips.  One is the mechanism of neuronal infection in vivo.  The authors acknowledge that neurons from dorsal root (DRG) or cranial nerve ganglia (CNG) can become infected with VZV either through their terminals in VZV-infected epidermis, or as a result of viremia, in which VZV-infected T-cells deliver virus directly to ganglionic neuronal cell bodies.  The current experiment only mimics infection through terminals.  No attempt was made to infect T cells and use these to infect neurons.  VZV is not secreted from infected cells as viable free virions and thus presumably spreads from cell to cell as a result of the fusion of VZV-infected cells with uninfected neighbors.  The authors, therefore, only partly mimic the in vivo infection of neurons. In the current experiments, moreover, the authors do not encounter the establishment of latency.  They speculate on why this might be so, but the current studies do not show the phenomenon of latency.  Because latency does not occur in the authors’ studies, therefore, the current experiments are artificial.  One might wonder whether the magnitude of viral infection is too high so that the viral load overwhelms the VZV-infected neurons and latency cannot occur.

The authors need to discuss how satellite cells in infected ganglia, which are not themselves infected, are stimulated to secrete INFa.  If infection reaches neuronal cell bodies in DRG and CNG by retrograde transport from VZV-infected axon terminals, and the infected neuronal cell bodies do not secrete viable virions, why would satellite cells secrete INFa?  It is likely that the infection of neurons in vivo would actually be latent and the satellite cells would, in fact, not secrete INFa.  The authors need to discuss the limitations of their model, which they claim mimics in vivo infection, but which, in major ways, does not do so.  The authors should also discuss how the establishment of latency would affect their conclusions, and perhaps more importantly, how their observations affect the establishment of viral latency.

The title of the current paper is “Activation of interferon-stimulated genes following varicella-zoster virus infection in a human iPSC-derived neuronal model depends on exogenous interferon-α”.  The authors show that the activation of interferon-stimulated genes in their model does, in fact, depend on exogenous interferon-α; however, the inference is that this phenomenon is relevant to VZV infection in vivo.  That relevance, however, has not been established by the current experiments.  The authors propose to do so in the future, but do not do so here.  The title thus is a bit strong and ought to be softened.  Alternatively, the co-culture experiments that the authors propose could be done, but softening the title seems easier. 

In summary, this otherwise outstanding paper needs to be modified by appropriate discussion of the limitations of the investigation and modification of the title.

Reviewer 2 Report

This is an important study on the innate immune response to varicella zoster virus (VZV) infection in human iPSC-derived neurons. VZV is an exclusively human virus, and a reproducible animal model is lacking. Therefore, neuronal culture model has been employed to address immune responses. They show that hiPSC-neurons grown in a microfluidic chamber do not produce interferon (IFN)- response upon axonal infection of the with cell associated VZV. They also demonstrate that exogenous addition of IFN-alpha limited virus spread and upregulated interferon-stimulated genes. The manuscript is written clearly and well-organized with appropriate controls. 

Addressing some minor comments will help the readers:

Line 79: they need to make it clear that that the reactivation experiments were done in cultured neurons invitro.

Line 126: include the names of the synthetic inducers, poly (dA:dT) and poly (I:C).

Lines 672-674: This statement does not seem valid since these authors (lines 516-519), also pretreated the cell body compartment with IFN-2α.

Lines 686-688: it should say “data not shown”.

Reviewer 3 Report

Review of Boeren et al “Activation of interferon-stimulated genes…”

Major comments:

This manuscript contains a great deal of difficult to do and painstaking experimental work. Unfortunately, I believe the experiments do not prove any important biological point. Even if it was proved, that iPSC-derived neurons don’t secrete IFN-alpha but do respond to it, this seems to be neither particularly surprising based on the known ability of VZV to suppress the response to IFN in other cells and not particularly relevant to the in-vivo situation in varicella disease. This is because peripheral ganglia contain satellite glia (that have been shown to be infected by VZV by the Arvin group in SCID-Hu mice) as well as fibroblasts, Schwann cells, endothelial cells. Any of these cells could secrete IFNs of all the classes and be responsible for upregulation of ISG in neurons in the disease state. Reductionist experiments are very important in science for determining mechanisms, but in this case, the lack of ganglionic components prevent conclusions from being made about in-vivo infection.

In addition, there are a number of serious technical issues that are present in the ms.

1)      The authors measure mRNA and say they are testing for interferons and interferon-stimulated genes. This is not strictly  true- it is possible to see increases in transcription that are not reflected by protein synthesis (and secretion in the case of many of these genes), and vice-versa, there could be increased release of immune-related molecules without measurable changes in mRNA. The text needs to be modified to say “mRNA of” or “message expression”, instead of expression at least once to indicate that no protein assays were performed in this study.

2)      A major control used in these experiments is the Sendai virus. In contrast to VZV, with a double-stranded DNA genome, SeV has a single-stranded RNA genome. Although there are some intracellular receptors that are in common to the intrinsic response to these two classes of viruses, there are many differences. Therefore, the control should have been another DNA virus, and the most logical one would be HSV, another human neurotropic alphaherpesvirus with a similar genome. Another control that really would add to the strength of conclusions reached would be UV inactivated VZV.

3)      Related to 2, the amount of SeV used may have been MUCH higher than that of VZV. The infection of the neurons was therefore likely to have been greater because of this, which could explain why SeV infection resulted in stimulation of some ISGs, while VZV infection did not.

4)      The introduction points out that VZV may reach the sensory ganglia in varicella disease either by axonal transport from the skin or via infected T-cells. Indeed, experiments with simian varicella virus have shown that VZV can be detected in ganglia before lesions are apparent in the skin, favoring the lymphocyte-vector for infection of neurons. This should be mentioned in the introduction and again in the discussion, especially in light of the claim in lines 671-672 that axonal infection is more like the pathway of infection in varicella disease than cell body infection (which could be via lymphocytes…).

5)      The quantification of the number of neurons infected by VZV in this ms is not described in sufficient detail, and the detail given, makes it problematic. The methods say that quantification of infection was performed by using ImageJ, without providing any detail how this was done, or demonstrating that it reflected the actual number of infected neurons.  A figure showing computer quantitation compared to human-counting of neurons is very much missing.

ORF23, as pointed out by the authors, is a capsid-associated gene and is expressed as puncta/dots at early stages of infection. In the legend to figure 3, the quantification of infection is given as “dots”. Since the GFP expression is punctate, dots do NOT reflect the number of neurons that were infected since multiple dots are present in each neuron. Indeed, the paper describing this virus (Lebrun 2014) is full of pictures of punctate GFP expression. The micrographs of the GFP expression in the figures in the paper itself are of too low a magnification to see how the GFP is distributed in the cells.

6)      Related to 4- I did not find mentioned anywhere in the manuscript how many neurons were seeded or completed the maturation process in each well of the Zona chambers. This is important in order to estimate the amount of VZV (or SeV) the neurons were exposed to. There is unlikely to be a 1:1 correspondence of axons traversing the channels to the number of neuronal cell bodies in the soma chamber, but at least an idea of the MOI given is quite important, since low doses of VZV (and HSV) are used in order to obtain a silent/latent infection in similar systems. The PFU of the VZV was determined on non-neuronal cells, so its impossible to know the quantitative relation of VZV/neurons from the methods as written.

7)      The authors did not demonstrate that the cultures consisted only of post-mitotic neurons, no staining for mitotically active cells was performed (or at least not mentioned). Beta-III tubulin is present in radial glia in the brain, and probably in neuronal precursors as well, and as such isn’t a very good marker. This is especially important in light of the fact that this study obtained productive infection with lysates, while other studies of VZV infections (and HSV infections) via axons lead to latent-like and not productive infections.

8)      The methods do not mention testing whether viruses were able to diffuse from the axonal to the cell body chambers or whether any precaution was taken to prevent this. Testing is simple using a fluorescent dye, and establishment of a volume/pressure gradient prevent flow from the axon to the cell body chambers is simple. Is it possible that virus placed in the axonal chamber diffused through the channels and directly infected the neuronal cell bodies? This would explain how productive infections were obtained in this study while others obtain latent infections. If the authors tested flowthrough, and took precautions to prevent it, it should be mentioned in the methods.

9)      VZV cell lysate (“cell-free”) preparations. A) Only looking for growth seems to not be adequate to know if there were living cells- were intact nuclei (live-cell Hoechst staining) and phase-bright cells present that did not divide? B) Lysate of infected cells contains intracellular and cell membranes with associated virus, organelles and of course, proteins and nucleic acids. It seems to me that some of the RT-PCR detections could have come from the input ARPE cytoplasm- without treatment with RNAase or centrifugation and demonstration that this lysate didn’t contain mRNA, it is difficult to to interpret results of infections with this material. This is especially true of the data shown in Figure 7, transcripts could have easily been carried over with the input ARPE lysate.

10)  If I understand the figure legends correctly,  many of experiments were performed more than once, but that all the material was pooled together for analyses. Traditional repeats of experiments would lend more confidence to the results presented. If I didn’t understand correctly, the text of the legends should be edited for clarity.

11)  The detection of IFNB1 after poly(da:dt)was at cycle 30+ Figure 3E, hardly a robust result to base conclusions upon.

Minor comments:

1)      It is important to point out, at least in the Methods, that the VZV used in the study is derived from the pOka strain and not a vaccine strain, since so many studies of VZV use vaccine-derived viruses.

2)      The line of IPSC-derived neural progenitors used by the authors was described by them as “previously characterized and validated”. What does validated mean here? Is this line of neural precursors used in other laboratories or available for distribution/purchase? In light of obtaining a productive infection where others see quiescent, the question of how representative this line/neural stem cells are is quite important.

3)      Why was multiplexing necessary for detecting VZV transcripts? If the neurons were productively infected a single pair of primers should suffice for detections

4)      I would have been happy to see a control for the peripherin staining (or at least mentioned) since chicken antibodies aren’t so specific in everyone’s hands.

5)      Line 378 An alternative explanation for increasing GFP expression from 3-7dpi to that of a lack of antiviral response given by the authors, is that infection by VZV in general, and probably via their axons specifically, is very asynchronous. It could have simply been that it took longer for some of the neurons to begin expressing sufficient GFP to be seen because it took more time for more virus to be transported to the somata and initiate a productive infection.

6)      The legend to figure 3D mentions measurement of VZV plaques, I did not find how this was performed in the methods section.

7)      The use of Log2 scales is not intuitive. The comparisons could have been done on a simple linear scale since the differences were relatively small and didn’t need a log10 scale.

8)      Line 685- The authors “did not detect any clear signs of quiescence in our experiments”. How was quiescence looked for? In-situ? PCR for DNA genomes? Does this mean that in every experiment productive infection by the lysates was obtained? That might be a better way to express this observation if this is the case.

9)      Figure 5 is very large and very busy. Could it be distilled into a table that is easier to read?

10)  Supplementary figure 1: Are the pictures of infected NSC or neurons? The legend says both, and very few axons are seen

11)  Line 27 Could the high dose have been toxic? Was this tested for?

Minor English issues

Line 123 need better word than “reflects”

Line 107 needs its grammar fixed

Line 225 21 DIV- is this 21 days after seeding in the Zona chambers? Then it really isn’t days in vitro, the precursors were in vitro and the iPSC were invitro. Perhaps 21 days of terminal differentiation?

Line 243 compartments, not compartment.
